# Attitudes toward Aging among College Students: Results from an Intergenerational Reminiscence Project

**DOI:** 10.3390/bs13070538

**Published:** 2023-06-28

**Authors:** Ling Xu, Noelle L. Fields, Jessica Cassidy, Kathryn M. Daniel, Daisha J. Cipher, Brooke A. Troutman

**Affiliations:** 1School of Social Work, University of Texas at Arlington, 501 W. Mitchell Street, Arlington, TX 76019, USA; noellefields@uta.edu (N.L.F.); jessica.cassidy@uta.edu (J.C.); 2College of Nursing and Health Innovation, University of Texas at Arlington, Arlington, TX 76019, USA; kdaniel@uta.edu (K.M.D.); cipher@uta.edu (D.J.C.); 3McDermott Library, United States Air Force Academy, Colorado Springs, CO 80840, USA; brooke.troutman@afacademy.af.edu

**Keywords:** attitudes toward aging, intergenerational connection, reminiscence, Fraboni Scale of Ageism

## Abstract

The detrimental effects of negative attitudes toward aging among younger adults extend to both older and young adults, highlighting the need for attention from academics, applied researchers, and practitioners. To improve college students’ attitudes toward aging, an intergenerational reminiscence intervention was conducted. College students, who were randomized to intervention or control groups and matched with older adults, made weekly phone calls to community-dwelling older adults with cognitive impairment for ten weeks. This study investigated whether college students improved their attitudes toward aging after participating in this project. A total of 64 college student participants completed the whole intervention and all data collection. The Fraboni Scale of Ageism was used to measure attitudes toward aging and administered at three time points (pre-, mid-, and post-test). Parametric and nonparametric tests were examined to understand changes over time, and post-hoc analyses were conducted to understand timepoints in which changes occurred. The results showed that both the intervention and control groups evidenced a decrease in the majority of the ageism scale, including statistical improvements in three specific negative items, which were “Seniors are stingy and hoard money”, “Seniors live in the past”, and “I prefer not to spend time with seniors”. Overall, the findings indicate that weekly engagement with older adults is promising in improving attitudes toward aging among college students. Implications for future research on intergenerational contacts to improve attitudes toward aging are discussed.

## 1. Introduction

Ageism is described as “negative or positive attitudes, prejudice and/or discrimination against aging people on the basis of their chronological age or on the basis of them being ‘old’ or ‘elderly’” [1]. Although there are few prevalence estimates of ageism, evidence gathered from an examination of 83,034 individuals across 57 countries found that 24% of participants had highly ageist attitudes [2]. Ageism can contribute to negative outcomes among older adults, as well as younger adults [3]. Among older adults, experiencing negative attitudes toward aging has been associated with poorer mental health outcomes, including depressive symptoms, anxiety, and generalized stress [4]. Experiences of ageism have also been associated with an increased likelihood of worsened physical outcomes, such as frailty among older adults [5]. Moreover, negative stereotypes suggesting that older adults are more forgetful than their younger counterparts may contribute to a greater likelihood of adults experiencing higher levels of anxiety of developing Alzheimer’s Disease and Related Dementias (ADRD) [6]. For younger adults, a growing body of literature indicates that younger adults’ negative attitudes toward aging can result in internalization of the projected stereotype leading to their own worsened health outcomes in older age, such as increased risk of cognitive decline, cardiovascular disease, and depression in later life [7,8]. The negative impact of ageism on older and younger generations warrants attention in academia and in practice.

Negative attitudes toward aging are common among young adults [9]. In a study exploring Twitter tweets posted by college students participating in a senior mentoring program, 12% of 354 students’ tweets were found to contain language-based age discrimination [10]. Ageist attitudes among younger adults’ stem from a variety of contextual factors, including racial and ethnic backgrounds [11,12]. Intrieri and Kurth [11] reported that non-Hispanic White students held more negative attitudes toward aging and higher levels of ageism compared with African American students. Liou et al. [12] examined U.S. and Taiwanese undergraduate college students’ attitudes toward aging through student drawings. The authors found that U.S. students largely perceived aging as physical changes related to functional limitations, represented by 40.8% of pictures displaying mobility aids, 22.6% displaying glasses, and 25.8% displaying long-term care facilities, compared with Taiwanese students of whom the majority perceived aging to be related to changes to physical appearances, represented by 64.1% of pictures displaying wrinkles and 59.2% displaying gray or thinning hair [12].

Differences in attitudes toward older adults with and without cognitive limitations have also been noted. Older adults, and especially persons with ADRD, are more likely to experience ageism due to associations of older age with cognitive limitations [13]. Younger adults may have greater fear of and pity for older adults with ADRD [14]. In addition, negative attitudes and stereotypes toward older adults increased during the COVID-19 pandemic [15]. Due to the heightened risk COVID-19 posed to older adults, older adults were often portrayed as vulnerable and frail, contributing to negative attitudes toward aging [16]. General ageism among college students and negative attitudes toward aging that increased during the COVID-19 pandemic suggest an urgent need to combat ageism toward older adults, specifically toward those with ADRD [15].

### 1.1. The Intergenerational Reminiscence Intervention

Intergenerational reminiscence interventions often connect younger generations with older adults and are typically characterized by older persons sharing their life experiences with their younger counterparts [17]. The literature has shown the positive impact of intergenerational learning experiences on students’ academic and personal development [18], especially among undergraduate and graduate students preparing for future work with older adults [19]. Studies further suggest that intergenerational learning experiences involving interactions with persons with ADRD also result in positive gains for college students in academic learning and attitudes toward older adults [20,21]. Involving college students in ADRD care can help younger generations gain a better historical and cultural understanding regarding the context of the older generation’s lived experiences, positively change their views of aging, and potentially increase their awareness of ADRD [22]. Moreover, interventions that incorporate components of reminiscence with an intergenerational approach have both social and psychological functions for persons with ADRD [23] and indicate potential for simultaneously benefiting the social and mental health of older adults and reducing ageism among younger generations. Intergenerational reminiscence incorporates components of reminiscence with an intergenerational approach and is characterized by older persons sharing their life experiences and life lessons to younger generations [17,24,25]. Though intergenerational reminiscence indicates potential to improve mental health outcomes and overall wellbeing among older adults, as well as to reduce ageism among younger generations, there is little existing evidence. Some efforts have attempted to utilize young volunteers as part of smaller scale intergenerational reminiscence studies [26,27]; however, the use of undergraduate and graduate students is thus far unexamined. As this approach may potentially offer a more sustainable and cost-effective intervention for improved overall wellbeing among community-dwelling older adults, particularly for those with ADRD, as well as reduced ageism for younger adults, further research is necessary to better understand intergenerational reminiscence with college student volunteers.

### 1.2. The Present Study

Given the prevalence of older adults with ADRD and the lack of evidence examining intergenerational reminiscence with college student volunteers, a telephone-based intergenerational reminiscence project was developed and conducted to connect college students with community-dwelling older adults who have cognitive impairments. The purpose of this project was to enhance the social and emotional wellbeing of both young and old participants, as well as to improve the knowledge and attitudes toward aging among young adults. This study reported changes among college students’ attitudes toward aging during and after participation in this project. Additionally, we investigated potential differences between participants in the intervention and control groups. Specifically, this study aimed to answer the following hypotheses:

(1) Young adult volunteers in both groups will have more positive attitudes toward aging after the intervention.

(2) Young adult volunteers in the intervention group will have more positive attitudes toward aging compared to those in the control group.

## 2. Methods

### 2.1. Samples

Community-dwelling older adults were recruited through Meals On Wheels in the Tarrant County Chapter in North Texas. The eligibility criteria for older adult participants included persons with the following characteristics: (1) aged 65 years or older; (2) had cognitive impairment (indicated by AD8 dementia screening interview scores ≥ 2) [28]; (3) had decisional capacity as measured by the San Diego Brief Assessment of Capacity to Consent (UBACC) [29]; (4) not participating in any other research protocol at the time of this study; and (5) could speak and understand English. The exclusion criteria for older adults were as follows: (1) participating in another trial; (2) had health issues that prevented them from participating in the study for 3 months; and (3) could not commit to being available for the full 10 weeks of intervention. Potential older adults were informed of the study through a case manager at Meals On Wheels. Those interested in participating gave permission to the case manager for the research team to call them. After the case manager informed the research team, two faculty members called potential participants to provide more information about the research protocol. If they remained interested in participating, their decision capacity was then assessed. Eligible participants who passed the capacity screening then provided verbal consent to participate using the university-approved informed consent form. Then, a follow-up phone call was scheduled to complete the baseline survey with a trained research assistant. Please refer to Appendix A for the demographic information of older adult participants who completed the intervention and data collection.

College student participants were recruited from a public university in North Texas. The inclusion criteria for college students included the following: (1) aged 18–30 years old; (2) currently enrolled as a student at the University of Texas at Arlington; and (3) could commit to being available for the full 10 weeks of the intervention. College student volunteers who had health issues or limited availability preventing their participation through the duration of the 10-week intervention were excluded from the study. Due to COVID-19 social distancing protocols, this study contacted and recruited participants by emailing student listservs, advertising on various university social media accounts, reaching out to students virtually in their online classes, and/or sharing flyers with student via Microsoft Teams. In addition, student volunteers were recruited by leveraging established relationships with various student organizations across multiple disciplines. Students with interest in participating contacted the research team directly. Informed consent and the baseline survey were sent to students through QuestionPro, a university-secured platform for data collection.

For this study, we only focused on the young adult participants. Over 100 student participants were matched with older adult partners at the beginning of the study; however; some of them were unable to begin the intervention or withdrew midway through the study. This was primarily due to either health issues affecting their senior partners or conflicts in their own schedules. A total of 64 students completed the intervention and all data collection. See Table 1 for the students’ demographic information.

### 2.2. Procedure

Following collection of baseline measures, older adult participants and student volunteers were randomly assigned to reminiscence or control groups. All student volunteers completed mandatory three-hour training (e.g., research protocol, weekly call guidance, communication with older adults with ADRD). Next, students were randomly matched with older adults. In the intervention group, students made weekly phone calls to older adult participants based on reminiscence. Specifically, older adults participated in a series of reminiscence sessions facilitated by trained young adult volunteers. These sessions took place on a weekly basis and lasted approximately 45–60 min each. Throughout the first six weeks of the intervention, a reminiscence interview was conducted with specific themes assigned to each week. The themes covered various aspects of the participants’ lives, including major turning points (week 1), family history (week 2), life and career accomplishments (week 3), history of relationships and emotions (week 4), experiences of stress (week 5), and the meaning and purpose of life (week 6). The last four weeks consisted of creating a product based on earlier reminiscence conversations. To ensure the quality of the reminiscence interviews, guidelines were provided to the young adult volunteers. These guidelines were developed based on the work of Watt and Cappeliez [30] and others [31], which focused on integrative and instrumental reminiscence therapies for older adults.

A sham control group was used in the present study as a way to control for placebo effects and to assess the specific effects of the intervention being studied. Compared to a no-treatment control group, sham control groups offer more advantages [32,33,34,35]. In the present study, the weekly calls in the control group had the same dosage and duration as the intervention group but centered around general topics related to health/wellbeing. Each week, a specific theme was discussed to explore its connection to health/wellbeing. The topics covered in the weekly calls were as follows: diet and health (week 1); exercise/activity and health (week 2); emotions and health (week 3); religious/spiritual practices and health (week 4); family/friend relationships and health (week 5); and social activities/engagement and health (week 6). During the final four weeks of the intervention, instead of focusing on digital storytelling, they collaborated with the older adults to create an unstructured, non-digital record of the social topics they discussed in the first week. This could take the form of a wellness scrapbook or journal, where the participants were asked to document and capture meaningful moments and experiences during their time together.

Weekly check-in calls (approximately 10–15 min each) were made by research assistants to the dyads to ensure the fidelity of the intervention, such as monitoring the content and quality of the phone calls made by the students. These calls included questions related to the study protocol (e.g., topics covered, time spent, challenges that occurred). If a student volunteer experienced emotional distress during the study, members of the research team included a clinical social worker and an RN (nurse practitioner) who offered consultation, and if necessary, referred student volunteers to university services as needed.

### 2.3. Measurement

Pre-, mid-, and post-survey questionnaires were administered through QuestionPro to measure students’ attitudes to aging using the Fraboni Scale of Ageism (FSA) [36]. The FSA was developed by Fraboni et al. [36] and revised by Rupp et al. [37]. The FSA scale includes 29 items, and each is measured by a 4-point Likert scale (1 = *strongly disagree* to 4 = *strongly agree*). Good psychometric properties of the FSA with high internal consistency and strong validity have been reported when tested in college students [11,38], among healthcare workers employed at a university and state hospital [39], as well as in other countries, such as among Chinese medical students [40].

### 2.4. Data Analysis

Descriptive analyses were first computed to provide overall means and frequencies on key outcome variables for participants. Shapiro–Wilk tests were computed to identify the normality of the study variable distributions. Due to non-normality, Friedman tests were computed to examine changes in each item of the FSA in the combined group (all participants), as well as within the intervention and control groups. Post-hoc analyses were conducted to understand changes in scores between assessments using Wilcoxon signed-rank tests. The extent and pattern of missing data were assessed with sensitivity analyses. The results revealed that 37.9% (*n* = 39) of the participants had at least one missing segment of follow-up data. The most salient pattern of missing data was the pattern of attrition. No other patterns were identifiable. Missing data imputation was performed using the expectation–maximization method to create an intent-to-treat imputed dataset [41,42]. However, the results of the analyses on the imputed dataset should be interpreted with caution due to the pattern of missing data that was identified. Analyses were performed on the complete-case data (*n* = 64) and the intent-to-treat imputed data (*n* = 103). The study alpha was set at 0.05. Analyses were conducted with SPSS 29 for Windows.

## 3. Results

Decrease in the mean scores of negative statements, as well as increase in the mean scores of positive statements (marked with *), indicated improvement in attitudes toward aging. The results indicated that student volunteers reported improvement in the majority of the FSA items after participating in the project, which was indicated by the mean changes at three time points (see Table 2). However, only three items were statistically significant at *p* < 0.05 level or marginally significant at *p* < 0.10 level (see Table 3). Reporting marginal significance in statistical analysis is important because marginally significant results can still provide valuable information, contribute to the body of evidence, and help refine theories or inform future research [43,44].

Significant improvements were observed for the combined groups in three items when examining the complete sample (*n* = 64). The first item was “Many old people are stingy and hoard their money and possessions”. The results showed this item significantly changed over three time points (χ^2^(2) = 8.70, *p* = 0.013). The post-hoc analyses shown in Table 3 indicated a significant decrease for both groups between pre- to post-test (mean difference = 0.27, SD = 0.761, *p* = 0.009) and a marginal significant decrease from mid- to post- test (mean difference = 0.14, SD = 0.560, *p* = 0.050) on this negative attitude item. The intervention group evidenced a marginally significant decrease in this item (χ^2^(2) = 5.77, *p* = 0.056) across the three time points. In addition, further analyses shown in Table 3 showed that a significant decrease occurred from pre- to post-test (*mean difference* = 0.36, *SD* = 0.638, *p* = 0.013) and a marginal significant decrease from mid to post-tests (*mean difference* = 0.20, *SD* = 0.577, *p* = 0.096). No significant decrease was observed in the control group when only examining the complete sample.

Significant decreases were also observed for the combined groups (*n* = 54) in the item “Many old people just live in the past” at three time points (χ^2^(2) = 15.44, *p* < 0.001). Post-hoc analyses in Table 3 also showed a significant decrease in this item between pre- and post-test (*mean difference* = 0.38, *SD* = 0.745, *p* < 0.001), pre- to mid-test (*mean difference* = 0.22, *SD* = 0.723, *p* = 0.020), and mid- to post- test (*mean difference* = 0.16, *SD* = 0.648, *p* = 0.059). For the intervention group, a marginally significant difference occurred at three time points (χ^2^(2) = 5.44, *p* = 0.066), and further analyses showed a decrease from pre- to post-test (*mean difference* = 0.40, *SD* = 0.645, *p* = 0.008) and mid- to post-test (*mean difference* = 0.20, *SD* = 0.577, *p* = 0.096). The control group also evidenced a significant decrease in this negative item at three time points (χ^2^(2) = 4.84, *p* = 0.001), namely between pre- and post-test (*mean difference* = 0.36, *SD* = 0.811, *p* = 0.01) and from pre- to mid-test (*mean difference* = 0.23, *SD* = 0.742, *p* = 0.061).

Lastly, “I personally would not want to spend much time with an old person” suggested a marginally significant improvement in the combined complete sample (χ^2^(2) = 4.82, *p* = 0.090) at three time points, with a decrease occurring from mid- to post-test (*mean difference* = 0.20, *SD* = 0.717, *p* = 0.028). For the control group, a marginally significant decrease was observed across three time points (χ^2^(2) = 5.20, *p* = 0.074) and further analyses showed that this decrease occurred from mid- to post-test (*mean difference* = 0.26, *SD* = 0.818, *p* = 0.058). No significant differences were found in the intervention group.

## 4. Discussion

This study aimed to improve college student volunteers’ attitudes toward aging through a telephone-based intervention delivered to community-dwelling older adults with cognitive impairment. This study contributed to the literature on this topic by focusing on older adults with cognitive impairment, integrating reminiscence or social wellness as weekly conversation topics, measuring outcomes at multiple time points, and using a telephone-based approach. By examining individual items on the Fraboni Ageism Scale, the findings suggest that an intergenerational connection between college students and older adults was associated with reduced negative attitudes toward aging among college students. Improved changes in the students’ attitudes toward aging reflect how intergenerational interventions may dispel stereotypes and increase younger generation’s positive perceptions of engaging with older adults.

First, this study connected older and young adults to engage in weekly conversations for 10 weeks. This approach was observed to improve college students’ negative attitudes toward aging. This corresponds to previous findings in the literature. For example, through a review of the literature, Blais et al. [18] reported that many studies indicate that college students improved their attitudes toward older adults through involvement in intergenerational programming. Establishing intergenerational contact and fostering relationships between older individuals and young adults can enable the younger generation to develop meaningful connections with older adults [20] and serve as effective measures to prevent the formation of stereotypes or negative attitudes, as well as challenge and dispel existing ones [20,22].

Second, this study focused on older adults with cognitive impairments and had similar findings to the literature on this topic. In particular, studies on intergenerational intervention connecting college students with older adults living with dementia have found that students’ attitudes toward older adults improved, and their knowledge and comfort regarding dementia increased [12,21,22]. The studies suggest that intergenerational interventions connecting college students with older adults can help younger generations gain a better historical and cultural understanding of the older generation’s lived experiences, positively change their views of aging, and potentially increase their awareness of ADRD disease.

Third, our study also contributes to the literature by adding reminiscences approach to the program and suggests similar findings may be achieved through intergenerational reminiscence interventions, especially for some specific ageism items, such as the very significant decrease in the ageism items “Older adults live in the past” and “Seniors are stingy and hoard money”. This finding confirms that reminiscence combined with an intergenerational approach may yield significant benefits for young adults. Through an intergenerational reminisce approach, older persons may pass on their life experience and life lessons to young generations, which may help change college students’ attitudes. Based on these findings, professionals and program managers may want to contemplate incorporating intergenerational weekly calls into their current programs or establishing a college reminiscence course that provides students with practical chances to connect with older adults via phone conversations to improve attitudes toward aging. To the best knowledge of the authors, there are only two studies [26,27] that have implemented intergenerational reminiscence with young volunteers. Though the study by Chung [26] showed positive appreciation of older persons by young volunteers, it did not measure attitudes toward aging. The work by Gaggioli [27] showed that children’s attitude toward older adults positively changed; however, the participants were primary elementary students and group reminiscence was used (1 senior and 6–8 children in a group).

Several limitations were observed in the present study. First, limitations of this study include an administrative error which resulted in the last four items of the survey not being administered to student volunteers. Thus, we could not report the sum scores or the sub scales. This limitation may affect the completeness and accuracy of the data analysis and interpretation. Second, although the sample’s racial and ethnic diversity is a strength of this study, the large representation of Asian or Asian American students, as well as Hispanic or Latino students, may have biased the results due to potentially stronger cultural values for respect toward older adults [11,21]. However, such a limitation correctly acknowledges that cultural values may have influenced the results, highlighting the need for further research with more diverse samples to generalize the findings. Third, we were unable to conduct a follow-up survey due to certain challenges. Some student participants had graduated, and the older adult participants had health complications or were no longer involved in the Meals on Wheels programs. As a result, it was difficult to reach out to these individuals for a follow-up assessment. Fourth, we did not include any covariates or confounding variables in the analyses. Future studies would strengthen the methodological rigor of the study by adding any potential covariates to test the changes in attitude toward aging over time.

Despite the limitations encountered, the current study has provided valuable insights into the attitudes toward aging among college students, highlighting key areas for future practice and research efforts. Firstly, the study’s findings indicate that both reminiscence and social wellness calls have the potential to positively influence young adults’ attitudes. Building upon these promising results, practitioners and program managers in the university or community could consider integrating intergenerational weekly calls into their existing programs, which can effectively promote positive attitudes toward aging among young adults. Secondly, when implementing an intergenerational program, it is recommended to develop a comprehensive working manual and provide relevant training, similar to the practices followed in this study. These resources can offer valuable information about the growing older population and provide guidance on effective and respectful communication strategies when interacting with older adults, which may help young adults shift their negative attitudes toward aging. Thirdly, given the common challenges associated with recruiting young adults, it would be beneficial to establish an elective reminiscence course in specific departments or across the university, such as a school of nursing or social work. By doing so, students enrolled in the course would have the opportunity to apply the principles of reminiscence they have acquired, as well as to put them into practice by engaging with older adults with ADRD within the community. Lastly, considering the study’s adaptation to a telephone-based intervention due to the COVID-19 pandemic, future research should explore the implementation of face-to-face intergenerational programs. By comparing the impact of these different approaches, researchers can determine which method offers a stronger influence on young adults’ attitudes toward aging.

## 5. Conclusions

The presence of negative attitudes toward aging among younger adults has significant repercussions for both older and young individuals, underscoring the importance of academic and practical attention to this area. This study successfully facilitated connections between young adults in college and older adults with cognitive impairment in the community. Overall, the findings of this study indicate that intergenerational intervention, which focused on weekly calls between college students and older adults, holds promise in improving attitudes toward aging.

Notably, the study observed an overall decrease in the majority of items of the FAS ageism scale, with three specific items showing statistically significant improvement: “Seniors are stingy and hoard money”; “Seniors live in the past”; and “I prefer not to spend time with seniors”. These findings provide valuable insights for practices aiming to enhance young adults’ attitudes toward aging. Based on these findings, professionals and program managers may want to contemplate incorporating intergenerational weekly calls into their current programs or establishing a college reminiscence course that provides students with practical chances to connect with older adults via phone conversations. Alternatively, in-person intergenerational engagement can also be implemented in lieu of weekly calls.

To advance the field, future research should expand upon this pilot study by developing a scalable and replicable model involving a larger and more diverse participant base. This expansion could involve factors such as different types of institutions (public vs. private colleges) and the inclusion of older adults from various agencies. It is crucial to continue research with multiple universities to gain a more comprehensive understanding of the intervention’s effectiveness. Ultimately, improving attitudes toward aging in college students is paramount, as it is essential to cultivate a workforce that is prepared to address the opportunities and challenges presented by an aging society.

## Figures and Tables

**Table 1 behavsci-13-00538-t001:** Demographics of young adult participants at baseline.

	Complete-Case Sample (*n* = 64)			Intent-to-Treat Imputed Sample (*n* = 103)		
	*n*	Mean (SD) [Range]	%	*n*	Mean (SD) [Range]	%
Intervention group	27		42.20%	48		46.6%
Age	64	23.4 (3.56) [18–30]			23.13 (3.18) [18–30]	
Gender_Female	50		78.10%	80		77.7%
Marital Status_Single	54		84.40%	88		85.4%
Degree in Progress						
Bachelor’s	37		57.80%			
Master’s or PhD	27		42.20%			
Race/Ethnicity						
Asian or Asian American	23		35.90%	37		35.9%
Hispanic or Latino	22		34.40%	39		37.9%
African American or Black	9		14.10%	11		10.7%
White and Others	10		15.60%	16		15.5%
Department of Study						
Social Work	21		32.80%	23		25.3%
Nursing	20		31.30%	37		40.7%
Others	23		35.90%	43		34.0%
Living Arrangement						
With parent	23		35.90%	36		35.0%
With roommate	19		29.70%	32		31.1%
Others	22		34.40%	35		33.9%

**Table 2 behavsci-13-00538-t002:** Percentage of respondents to each item of ageism scale.

Ageism Scale	Pre-Test	Mid-Test	Post-Test	*p* ^1^	*p* ^2^
	*Mean* (*SD*) ^1^	*Mean* (*SD*) ^1^	*Mean* (*SD*) ^1^		
1. Teenage suicide is more tragic	2.08 (0.719)	2.11 (0.838)	2.09 (0.849)	0.975	0.471
2. Special clubs should be set for seniors	3.14 (0.710)	3.08 (0.803)	2.98 (0.845)	0.420	0.109
3. Seniors are stingy and hoard money	1.84 (0.695)	1.72 (0.548)	1.58 (0.686)	0.013	<0.001
4. Seniors are not making new friends	2.11 (0.737)	1.89 (0.799)	1.84 (0.739)	0.147	0.003
5. Seniors live in the past	2.11 (0.645)	1.89 (0.669)	1.73 (0.623)	<0.001	<0.001
6. I avoid eye contact with seniors	1.52 (0.690)	1.55 (0.665)	1.45 (0.589)	0.401	0.002
7. I don’t like conversation with seniors	1.38 (0.519)	1.39 (0.553)	1.39 (0.523)	0.985	0.108
* 8. Seniors deserve the same rights	3.67 (0.592)	3.64 (0.627)	3.59 (0.660)	0.363	0.022
9. Complex conversations are not for seniors	1.45 (0.711)	1.50 (0.797)	1.66 (0.946)	0.311	0.016
10. Depressed around elderly people	1.83 (0.703)	1.70 (0.749)	1.69 (0.639)	0.246	0.687
11. Seniors should find friends of their own age	2.03 (0.689)	1.91 (0.750)	1.97 (0.734)	0.124	0.040
* 12. Seniors are welcome at social gatherings	3.47 (0.563)	3.55 (0.641)	3.50 (0.563)	0.162	0.028
13. I prefer not to attend senior clubs	1.77 (0.660)	1.77 (0.636)	1.64 (0.675)	0.102	0.010
* 14. Seniors can be very creative	3.52 (0.534)	3.55 (0.589)	3.42 (0.686)	0.368	0.078
15. I prefer not to spend time with seniors	1.59 (0.610)	1.66 (0.672)	1.45 (0.589)	0.090	0.003
16. Seniors should not renew their driver’s license	2.08 (0.543)	2.09 (0.771)	2.05 (0.744)	0.928	0.173
17. Seniors should not use community sports facility	1.47 (0.503)	1.45 (0.665)	1.48 (0.642)	0.706	0.009
18. Seniors should not be trusted to care for infants	1.88 (0.577)	1.81 (0.664)	1.73 (0.672)	0.157	0.011
19. Seniors are happiest with people their own age	2.42 (0.612)	2.27 (0.597)	2.34 (0.695)	0.209	0.087
20. It is better seniors live where they won’t bother anyone	1.47 (0.689)	1.39 (0.581)	1.34 (0.479)	0.522	0.190
* 21. The company of most seniors is enjoyable	3.39 (0.633)	3.41 (0.610)	3.41 (0.660)	0.878	0.138
* 22. It is sad to hear about the plight of the old	3.44 (0.588)	3.44 (0.639)	3.36 (0.601)	0.660	0.078
* 23. Seniors should speak out politically	3.25 (0.591)	3.23 (0.636)	3.20 (0.694)	0.888	0.089
* 24. Most seniors are interesting	3.48 (0.563)	3.45 (0.641)	3.48 (0.617)	0.965	0.030
25. Seniors have poor hygiene	1.94 (0.614)	1.73 (0.648)	1.80 (0.603)	0.104	0.047

* Positive aging. ^1^ Results of Friedman tests performed on complete sample *n* = 64. ^2^ Results of Friedman tests performed on imputed sample *n* = 103.

**Table 3 behavsci-13-00538-t003:** Observed differences in three ageism items.

Significant Items from Ageism Scale	Both Groups			Intervention			Control		
	Mean Difference (SD) ^1^	*p* ^1^	*p* ^2^	Mean Difference (SD) ^1^	*p* ^1^	*p* ^2^	Mean Difference (SD) ^1^	*p* ^1^	*p* ^2^
3. Seniors are stingy and hoard money									
Pre- to Mid-test	0.16 (0.766)	0.223	0.078	0.16 (0.688)	0.248	0.384	0.13 (0.821)	0.513	0.114
Mid- to Post-test	0.14 (0.560)	0.050	<0.001	0.20 (0.577)	0.096	0.009	0.10 (0.552)	0.248	0.027
Pre- to Post- test	0.27 (0.761)	0.009	<0.001	0.36 (0.638)	0.013	0.010	0.21 (0.833)	0.135	0.009
Pre-, Mid-, Post-test (Friedman test)	χ^2^(2) = 8.70	0.0130	<0.001	χ^2^(2) = 5.77	0.056	0.003	χ^2^(2) = 3.55	0.170	0.002
5. Seniors live in the past									
Pre- to Mid-test	0.22 (0.723)	0.02	0.033	0.20 (0.707)	0.166	0.158	0.23 (0.742)	0.061	0.098
Mid- to Post-test	0.16 (0.648)	0.059	0.001	0.20 (0.577)	0.096	0.033	0.13 (0.695)	0.251	0.020
Pre- to Post-test	0.38 (0.745)	<0.001	<0.001	0.40 (0.645)	0.008	0.016	0.36 (0.811)	0.01	0.010
Pre-, Mid-, Post-test (Friedman test)	χ^2^(2) = 15.44	<0.001	<0.001	χ^2^(2) = 5.44	0.066	0.002	χ^2^(2) = 4.84	0.001	0.006
15. I prefer not to spend time with seniors									
Pre- to Mid-test	−0.06 (794)	0.414	0.124	0.00 (0.500)	1.00	0.226	−0.10 (0.940)	0.382	0.292
Mid- to Post-test	0.20 (0.717)	0.028	<0.001	0.12 (0.526)	0.257	0.022	0.26 (0.818)	0.058	0.008
Pre- to Post-test	0.14 (0.687)	0.115	0.179	0.12 (0.526)	0.257	0.803	0.15 (0.779)	0.244	0.142
Pre-, Mid-, Post-test (Friedman test)	χ^2^(2) = 4.82	0.090	0.003	χ^2^(2) = 1.40	0.497	0.087	χ^2^(2) = 5.20	0.074	0.017

^1^ Results of nonparametric comparison tests performed on complete sample *n* = 64. ^2^ Results of nonparametric comparison tests performed on imputed sample *n* = 103.

## Data Availability

The data presented in this study are available on request from the corresponding author.

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
