# Peer review of "Attitudes toward Aging among College Students: Results from an Intergenerational Reminiscence Project"

_behavsci, 2023, doi:10.3390/bs13070538_

Round 1
Reviewer 1 Report
Title and abstract
The abstract provides a concise overview of the study, including the intervention design, participant information, measurement tool, and key findings. It effectively communicates the study's purpose and outcomes. However, there are a few areas that could be improved to enhance the clarity and comprehensiveness of the abstract.
Firstly, it would be beneficial to include a brief statement about the significance of improving college students' attitudes toward aging. This would help readers understand the broader implications of the research and its potential relevance to the field. Additionally, the abstract could provide a concise summary of the specific findings related to the three items that showed statistical differences between the intervention and control groups.
Introduction
The introduction provides a concise overview of the research topic and identifies the gaps in the existing literature regarding intergenerational reminiscence interventions involving college students and older adults with cognitive impairment. It effectively sets the context for the study and highlights the importance of addressing these gaps. However, there are a few areas where improvements could be made to enhance the clarity and coherence of the introduction.
Firstly, it would be beneficial to provide a brief summary of the existing evidence supporting the positive outcomes of intergenerational reminiscence interventions for mental health, well-being, and perceptions of aging. This would help establish the rationale for conducting the current study and emphasize the potential significance of the findings.
Secondly, while the introduction identifies the specific gaps in the literature, it does not explicitly state the research objectives or research questions. It would be helpful to clearly state the aims of the study, such as investigating the changes in college students' attitudes toward aging and comparing the intervention and control groups.
Finally, the introduction could benefit from a more seamless flow between the gap identification and the proposed study. Currently, there is a slight disconnection between the gap identification and the specific intervention and study design chosen to address those gaps. Providing a clearer transition between these sections would improve the overall coherence of the introduction.
Materials and Methods
2.1. Samples
The description of the sample recruitment process for older adults and college students is brief but provides relevant information on the source and eligibility criteria for each group. However, additional details could be included to enhance the clarity and transparency of the recruitment process. Specifically, information about the inclusion and exclusion criteria for both groups, such as age range or specific cognitive impairment criteria for older adults, would help readers better understand the characteristics of the participants.
2.2. Procedure:
The procedure section provides a clear outline of the steps involved in the study, including random assignment, training for student volunteers, and the content of the intervention and control group activities. However, there are a few areas where further information could be beneficial. First, details regarding the duration and frequency of the intervention and control group activities would help in understanding the intensity and dosage of the intervention. Second, it would be useful to provide information on the measures taken to ensure the fidelity of the intervention, such as monitoring the content and quality of the phone calls made by the students. Including these details would strengthen the methodology section and enhance the rigor of the study.
2.3. Measurement:
The measurement section provides relevant information about the instrument used to measure students' attitudes toward aging. However, additional details could be included to enhance the readers' understanding. Specifically, information on the psychometric properties of the Fraboni Scale of Ageism, such as reliability and validity, would strengthen the methodological rigor of the study. Additionally, it would be helpful to briefly describe the three subscales of the FSA and provide examples of the types of items included in each subscale. This information would aid readers in comprehending the specific aspects of attitudes toward aging that the scale assesses.
2.4. Data Analysis:
The data analysis section provides a clear overview of the statistical procedures employed to analyze the data. It mentions the use of descriptive and exploratory analyses, as well as the specific statistical tests utilized, such as Friedman tests, independent t-tests, and Wilcoxon signed-rank tests. However, further details could be included to enhance the transparency of the analyses. Specifically, providing information on how missing data was handled and any potential covariates or confounding variables considered in the analyses would strengthen the methodological rigor of the study.
Results
The presentation of the results provides a concise summary of the findings, highlighting the overall improvement in FSA items for student volunteers after participating in the project.
The presentation of the significant improvements in the FSA items is clear, with the relevant statistical tests and p-values provided. However, further information regarding the direction of the changes (i.e., whether the attitudes became more positive or negative) would be beneficial in understanding the practical significance of the findings.
Lastly, the mention of "marginally significant" changes in some items suggests a potential trend but falls short of achieving statistical significance. While these results may still be of interest, providing additional context or discussing possible reasons for the marginal significance would strengthen the interpretation of the findings.
Discussion
The discussion section provides a comprehensive overview of the study's findings and their implications. The authors highlight the contribution of their study to the literature by focusing on older adults with cognitive impairment, integrating reminiscence or social wellness as conversation topics, measuring outcomes at multiple time points, and using a telephone-based approach. The findings suggest that the intergenerational connection between college students and older adults led to reduced negative attitudes toward aging among the students, reflecting the potential of intergenerational interventions to dispel stereotypes and foster positive perceptions of engaging with older adults.
The authors support their findings by referring to the literature, specifically citing a study by Blais et al. [8] that reported similar improvements in college students' attitudes toward older adults through intergenerational programming. They also highlight a study that focused on intergenerational interventions connecting college students with older adults living with dementia, which found improvements in attitudes, knowledge, and comfort regarding dementia [9]. By adding a reminiscence approach to their program, the authors suggest that similar findings can be achieved, particularly in addressing specific ageism items such as the perception that "older adults live in the past."
However, the discussion section could be strengthened by providing more in-depth interpretation and analysis of the findings. It would be beneficial to discuss the potential underlying mechanisms through which the intergenerational intervention influenced the students' attitudes toward aging. Additionally, the authors could explore the practical implications of their findings and discuss how these results can be applied in real-world settings or inform future interventions.
Limitations of the study are acknowledged, including an administrative error that resulted in the omission of the last four survey items for student volunteers. This limitation should be addressed, as it affects the completeness and accuracy of the data analysis and interpretation. Furthermore, the authors mention the potential bias introduced by the racial and ethnic composition of the sample, with a large representation of Asian or Asian American and Hispanic or Latino students. They correctly acknowledge that cultural values may have influenced the results, highlighting the need for further research with more diverse samples to generalize the findings.
Conclusion:
The conclusion provides a concise summary of the study's key findings and emphasizes the potential of intergenerational interventions, particularly weekly calls between college students and older adults, in improving attitudes toward aging. The authors suggest the importance of continued research involving multiple universities and highlight the significance of improving attitudes toward aging in college students to address the challenges and opportunities presented by an aging society.
Overall, the discussion and conclusion sections effectively convey the study's findings, their implications, and the need for further research. However, there are a few aspects that could be improved:
- In the discussion section, it would be beneficial to provide more in-depth interpretation and analysis of the findings. This could include exploring the underlying mechanisms through which the intergenerational intervention influenced the students' attitudes toward aging. Providing more detailed explanations and insights would enhance the understanding of the results.
- The practical implications of the findings could be discussed further. How can the results be applied in real-world settings? What are the potential implications for designing future interventions or programs? Adding a discussion of the practical implications would enhance the relevance and applicability of the study.
- While the limitations of the study are acknowledged, it would be helpful to address them in more detail. Specifically, the administrative error resulting in the omission of four survey items for student volunteers should be discussed thoroughly, including the potential impact on the overall findings and conclusions. Additionally, discussing how these limitations could be addressed or mitigated in future research would strengthen the article.
- The conclusion section could be expanded to provide a more comprehensive summary of the study's key findings and their significance. This could include a brief recap of the specific improvements in attitudes toward aging observed in the study and their implications for addressing the challenges and opportunities of an aging society. Providing a stronger emphasis on the key takeaways would leave a lasting impression on the readers.
By addressing these areas of improvement, the discussion and conclusion sections can be further strengthened, providing a more robust and impactful representation of the study's findings and their implications.References
References
The work relies on few bibliographic references, and these are not current.
Author Response
Comments and Suggestions for Authors
Title and abstract
The abstract provides a concise overview of the study, including the intervention design, participant information, measurement tool, and key findings. It effectively communicates the study's purpose and outcomes. However, there are a few areas that could be improved to enhance the clarity and comprehensiveness of the abstract.
- Firstly, it would be beneficial to include a brief statement about the significance of improving college students' attitudes toward aging. This would help readers understand the broader implications of the research and its potential relevance to the field. Additionally, the abstract could provide a concise summary of the specific findings related to the three items that showed statistical differences between the intervention and control groups.
Response: Thanks reviewer for this comment. We added the significance of improving college students’ attitudes towards aging in the beginning of the abstract. We also added a summary of the three specific items.
Introduction
The introduction provides a concise overview of the research topic and identifies the gaps in the existing literature regarding intergenerational reminiscence interventions involving college students and older adults with cognitive impairment. It effectively sets the context for the study and highlights the importance of addressing these gaps. However, there are a few areas where improvements could be made to enhance the clarity and coherence of the introduction.
- Firstly, it would be beneficial to provide a brief summary of the existing evidence supporting the positive outcomes of intergenerational reminiscence interventions for mental health, well-being, and perceptions of aging. This would help establish the rationale for conducting the current study and emphasize the potential significance of the findings.
Response: We appreciate this comment and agreed with the reviewer about the benefits of rich literature. We added information about ageism among older adults and its detrimental effects on both young and older adults. We also added summary of existing literature on intergenerational reminiscence intervention on pages 1-3.
- Secondly, while the introduction identifies the specific gaps in the literature, it does not explicitly state the research objectives or research questions. It would be helpful to clearly state the aims of the study, such as investigating the changes in college students' attitudes toward aging and comparing the intervention and control groups.
Response: Thanks reviewer for this comment. We added study purpose and specific aims on page 3 in “The present study” section.
- Finally, the introduction could benefit from a more seamless flow between the gap identification and the proposed study. Currently, there is a slight disconnection between the gap identification and the specific intervention and study design chosen to address those gaps. Providing a clearer transition between these sections would improve the overall coherence of the introduction.
Response: We appreciate the reviewer's comment and have worked to improve the transition and coherence of the introduction. Specifically, we have included a summary of the research gaps at the end of the "intergenerational reminiscence intervention" section and at the beginning of "The present study" section. By doing so, we believe that the revised introduction adequately highlights the research gaps and demonstrates how our study addresses them. Additionally, we have focused on creating clearer transitions between paragraphs in the revised manuscript. Thank you for your valuable feedback.
Materials and Methods
2.1. Samples
- The description of the sample recruitment process for older adults and college students is brief but provides relevant information on the source and eligibility criteria for each group. However, additional details could be included to enhance the clarity and transparency of the recruitment process. Specifically, information about the inclusion and exclusion criteria for both groups, such as age range or specific cognitive impairment criteria for older adults, would help readers better understand the characteristics of the participants.
Response: We added two paragraphs under “Samples” and described the older adult and young adult participants respectively. We also added a supplemental table to describe the demographic of older adult participants as suggested by reviewer 3.
2.2. Procedure:
- The procedure section provides a clear outline of the steps involved in the study, including random assignment, training for student volunteers, and the content of the intervention and control group activities. However, there are a few areas where further information could be beneficial. First, details regarding the duration and frequency of the intervention and control group activities would help in understanding the intensity and dosage of the intervention. Second, it would be useful to provide information on the measures taken to ensure the fidelity of the intervention, such as monitoring the content and quality of the phone calls made by the students. Including these details would strengthen the methodology section and enhance the rigor of the study.
Response: We appreciate the comments from the reviewer. We added detailed information in two separate paragraphs on page 5 about the intervention procedures for the intervention and control group respectively. In addition, we added another paragraph about our weekly calls for the purpose of fidelity checks on pages 5-6.
2.3. Measurement:
- The measurement section provides relevant information about the instrument used to measure students' attitudes toward aging. However, additional details could be included to enhance the readers' understanding. Specifically, information on the psychometric properties of the Fraboni Scale of Ageism, such as reliability and validity, would strengthen the methodological rigor of the study. Additionally, it would be helpful to briefly describe the three subscales of the FSA and provide examples of the types of items included in each subscale. This information would aid readers in comprehending the specific aspects of attitudes toward aging that the scale assesses.
Response: We value the reviewer's feedback and have made updates to provide additional details regarding the psychometric properties of the FSA scale, highlighting its reliability and validity across various populations. However, to maintain consistency and avoid overwhelming the readers, we have decided not to include further information on the three subscales of the FSA since they were not analysed in the present study.
2.4. Data Analysis:
- The data analysis section provides a clear overview of the statistical procedures employed to analyze the data. It mentions the use of descriptive and exploratory analyses, as well as the specific statistical tests utilized, such as Friedman tests, independent t-tests, and Wilcoxon signed-rank tests. However, further details could be included to enhance the transparency of the analyses. Specifically, providing information on how missing data was handled and any potential covariates or confounding variables considered in the analyses would strengthen the methodological rigor of the study.
Response: We added more information about the analyses on page 6, including how to deal with missing data. Furthermore, no covariates were included in the analyses, and this limitation has been addressed in the discussion section (see page 10).
Results
The presentation of the results provides a concise summary of the findings, highlighting the overall improvement in FSA items for student volunteers after participating in the project.
- The presentation of the significant improvements in the FSA items is clear, with the relevant statistical tests and p-values provided. However, further information regarding the direction of the changes (i.e., whether the attitudes became more positive or negative) would be beneficial in understanding the practical significance of the findings.
Response: Thank the reviewer for this comment. We added a sentence on page 6 “Decrease in the mean scores of negative statements as well as increase in the mean scores of positive statements (marked with *) indicated improvement in ageism item”. We also added the directions of the changes in the results part.
- Lastly, the mention of "marginally significant" changes in some items suggests a potential trend but falls short of achieving statistical significance. While these results may still be of interest, providing additional context or discussing possible reasons for the marginal significance would strengthen the interpretation of the findings.
Response: Thanks for this comment. We added the benefits of reporting marginal significance in statistical analysis on page 6.
Discussion
The discussion section provides a comprehensive overview of the study's findings and their implications. The authors highlight the contribution of their study to the literature by focusing on older adults with cognitive impairment, integrating reminiscence or social wellness as conversation topics, measuring outcomes at multiple time points, and using a telephone-based approach. The findings suggest that the intergenerational connection between college students and older adults led to reduced negative attitudes toward aging among the students, reflecting the potential of intergenerational interventions to dispel stereotypes and foster positive perceptions of engaging with older adults.
The authors support their findings by referring to the literature, specifically citing a study by Blais et al. [8] that reported similar improvements in college students' attitudes toward older adults through intergenerational programming. They also highlight a study that focused on intergenerational interventions connecting college students with older adults living with dementia, which found improvements in attitudes, knowledge, and comfort regarding dementia [9]. By adding a reminiscence approach to their program, the authors suggest that similar findings can be achieved, particularly in addressing specific ageism items such as the perception that "older adults live in the past."
- However, the discussion section could be strengthened by providing more in-depth interpretation and analysis of the findings. It would be beneficial to discuss the potential underlying mechanisms through which the intergenerational intervention influenced the students' attitudes toward aging. Additionally, the authors could explore the practical implications of their findings and discuss how these results can be applied in real-world settings or inform future interventions.
Response: We appreciate the comments from the reviewer. We added more discussion regarding the mechanisms through which the intervention influenced students’ attitudes toward aging on pages 8-9. We also added the potential implications for future practices and interventions on page 10.
- Limitations of the study are acknowledged, including an administrative error that resulted in the omission of the last four survey items for student volunteers. This limitation should be addressed, as it affects the completeness and accuracy of the data analysis and interpretation. Furthermore, the authors mention the potential bias introduced by the racial and ethnic composition of the sample, with a large representation of Asian or Asian American and Hispanic or Latino students. They correctly acknowledge that cultural values may have influenced the results, highlighting the need for further research with more diverse samples to generalize the findings.
Response: We appreciate this comment regarding the study limitations. As suggested, we added more information about the limitation of administrative error and bias of racial/ethnic composition of the sample. We also added two more limitations of the present study on pages 9-10, including not conducting following up surveys and no covariates included in the analysis.
Conclusion:
The conclusion provides a concise summary of the study's key findings and emphasizes the potential of intergenerational interventions, particularly weekly calls between college students and older adults, in improving attitudes toward aging. The authors suggest the importance of continued research involving multiple universities and highlight the significance of improving attitudes toward aging in college students to address the challenges and opportunities presented by an aging society.
Overall, the discussion and conclusion sections effectively convey the study's findings, their implications, and the need for further research. However, there are a few aspects that could be improved:
- In the discussion section, it would be beneficial to provide more in-depth interpretation and analysis of the findings. This could include exploring the underlying mechanisms through which the intergenerational intervention influenced the students' attitudes toward aging. Providing more detailed explanations and insights would enhance the understanding of the results.
Response: Thanks for this comment. We expanded the discussion part with 4 paragraphs to summarize and interpret the findings before we talk about study limitations (pages 8-9). Please also see response to comment #11 above.
- The practical implications of the findings could be discussed further. How can the results be applied in real-world settings? What are the potential implications for designing future interventions or programs? Adding a discussion of the practical implications would enhance the relevance and applicability of the study.
Response: We also added one paragraph about study implications on page 10. Please also see response to comment #11 above.
- While the limitations of the study are acknowledged, it would be helpful to address them in more detail. Specifically, the administrative error resulting in the omission of four survey items for student volunteers should be discussed thoroughly, including the potential impact on the overall findings and conclusions. Additionally, discussing how these limitations could be addressed or mitigated in future research would strengthen the article.
Response: We revised the limitation section according. Please also see response to #12 above.
- The conclusion section could be expanded to provide a more comprehensive summary of the study's key findings and their significance. This could include a brief recap of the specific improvements in attitudes toward aging observed in the study and their implications for addressing the challenges and opportunities of an aging society. Providing a stronger emphasis on the key takeaways would leave a lasting impression on the readers.
Response: We appreciate this comment from reviewer. We expanded the conclusion section to 3 paragraphs in the revised manuscript to include summary of the key findings and their significance, program implications, and recommendations for future studies.
By addressing these areas of improvement, the discussion and conclusion sections can be further strengthened, providing a more robust and impactful representation of the study's findings and their implications.
References
- The work relies on few bibliographic references, and these are not current.
Response: Thank you for this comment. We expanded the main text and therefore added more recent references in the revised manuscript.
Reviewer 2 Report
Your study is well documented, and your statistics are well organized. A few additions will make the manuscript more useful to the field. There is always a balancing act between staying brief and giving enough information for the reader to understand the project and how to interpret the results. Below are some suggestions about how to clarify the study.
A brief discussion on why 10 weeks was chosen as the intervention period would be helpful.
How long after the last weekly call did the post intervention data get collected? What does that time interval allow you to say about what it would take to make these changes permanent?
Table 2 shows the demographics of the students. A table (or an appendix) of the demographics of the older adult participants will make this study more useful to future researchers, as we work to include equity and access (and, yes, address racism) in our research programming.
You indicate that the students signed consent papers. What about the older adults? Did the older adults receive any training?
Why did you not use another control group where no training and no interactions took place? This would have allowed you to address the impact of the 3-hour training as well as the responsibility of making the weekly calls.
You say the students were volunteers. Where did they come from? Were they college students in a particular field of study? What impact might their educational interests have on the study and its results? Did you have any dropouts (of either students or older adults)? Did you monitor the calls? Were the calls scripted? Did the older adults (with cognitive impairments) have caregivers who monitored the calls? If so, what did these caregivers think of the study?
I see no comments or observations from the researchers or from the older adult participants about the students level of engagement or attitudes yet both groups have valuable insights.
There are minor errors in the English: such as in line 44 where the work connecting should be connected but these could be caught by the authors when they reread the manuscript and make changes.
Author Response
Comments and Suggestions for Authors
Your study is well documented, and your statistics are well organized. A few additions will make the manuscript more useful to the field. There is always a balancing act between staying brief and giving enough information for the reader to understand the project and how to interpret the results. Below are some suggestions about how to clarify the study.
- A brief discussion on why 10 weeks was chosen as the intervention period would be helpful.
Response: Thank you for this comment. The intervention duration of 10 weeks was determined by the literature on reminiscence therapy (6 weeks) and digital storytelling (4 weeks). We have elaborated on components of the intervention provided each week.
- How long after the last weekly call did the post intervention data get collected? What does that time interval allow you to say about what it would take to make these changes permanent?
Response: During the middle of the intervention, specifically in week 7 and within 1-2 weeks after the final week of the intervention, the participants were asked to complete surveys. This time interval was selected to provide ample opportunity for the participants to reflect on any changes in their attitudes towards aging that may have occurred as a result of the intervention.
However, we were unable to conduct a follow-up survey due to certain challenges and have detailed this in our edits. Some student participants had graduated, and the older adult participants were either hospitalized or no longer involved in the Meals on Wheels (MOW) programs. As a result, it was difficult to reach out to these individuals for a follow-up assessment. We added this limitation regarding the lack of a follow-up test in the discussion section on page 10.
- Table 2 shows the demographics of the students. A table (or an appendix) of the demographics of the older adult participants will make this study more useful to future researchers, as we work to include equity and access (and, yes, address racism) in our research programming.
Response: Thank you for this comment. We added a supplementary table to show the demographics of the older adult participants.
- You indicate that the students signed consent papers. What about the older adults? Did the older adults receive any training?
Response: Thank you for your questions. Yes, all participants including both young and older adults signed the consent form. We added this information on page 4 in the revised manuscript. The older adults did not receive training.
- Why did you not use another control group where no training and no interactions took place? This would have allowed you to address the impact of the 3-hour training as well as the responsibility of making the weekly calls.
Response: It would be ideal to have three arm randomized control trial (RCT). With the limited time frame and budget, we decided to conduct two arm RCT intervention with sham group rather than no treatment as the control group. Sham control groups are commonly used in intervention studies, particularly in clinical trials, as a way to control for placebo effects and to assess the specific effects of the intervention being studied. While using a no-treatment control group is also an option in some cases, sham control groups offer some advantages (Cigna, 2022; Hrong et al., 2003; Miller & Kaptchunk, 2004; Sutherland, 2007) as shown below. We added the rational of using sham group on page 5.
First, sham control group may provide placebo effects where participants believe they are receiving a treatment even if the treatment itself has no therapeutic value. Researchers can thus assess the specific effects of the intervention beyond any placebo response. This helps to determine the true efficacy of the intervention.
Second, sham control group has more power in blinding. In a sham control group, both the participants and the majority of researchers are typically blinded, meaning they do not know whether they are receiving or administering the active intervention or the sham treatment. This helps to minimize bias and ensures that any observed effects are more likely due to the intervention itself, rather than the participants' or researchers' expectations.
Third, having sham group is of ethical consideration because no-treatment control group may be considered as unethical when there is already evidence supporting the efficacy of the intervention (e.g., weekly phone call to older adults). It may be deemed inappropriate to withhold a potentially beneficial treatment from participants in a no-treatment group when an active intervention is available. In such cases, a sham control group allows for the ethical provision of an active intervention while still assessing its effects accurately.
- Placebo and sham treatment. 2022. https://www.cigna.com/knowledge-center/hw/placebo-and-sham-treatment-stp1708
- Horng S, Miller FG. Ethical framework for the use of sham procedures in clinical trials. Critical care medicine. 2003;31(3):S126-130.
- Miller FG, Kaptchuk TJ. Sham procedures and the ethics of clinical trials. J R Soc Med. 2004;97(12):576-578. doi:10.1177/014107680409701205
- Sutherland ER. Sham procedure versus usual care as the control in clinical trials of devices: which is better? Proceedings of the American Thoracic Society. 2007;4(7):574-576.
- You say the students were volunteers. Where did they come from? Were they college students in a particular field of study? What impact might their educational interests have on the study and its results? Did you have any dropouts (of either students or older adults)? Did you monitor the calls? Were the calls scripted? Did the older adults (with cognitive impairments) have caregivers who monitored the calls? If so, what did these caregivers think of the study?
Response: Thank you for bringing up this question. We added a paragraph about young adult student participants on pages 3-4. Furthermore, on page 4, we have included the attrition rates to provide transparency regarding participant retention. Additionally, on page 5, a brief paragraph has been added to explain the weekly calls to assess fidelity. Since all participants were required to pass the decision-making capacity assessment as an eligibility criterion, the older adults did not require caregivers to monitor the calls. The calls were not scripted but students received training prior to the calls. The training manual did provide suggested prompts to help guide the conversations.
- I see no comments or observations from the researchers or from the older adult participants about the students level of engagement or attitudes yet both groups have valuable insights.
Response: We greatly appreciate your comment. We agree with the reviewer's comment that the insights from both researchers and older adult participants would be valuable in the context of this study. However, the primary focus of this particular paper is solely on the self-feedback of students regarding their attitudes toward aging, as reflected in their surveys. We are currently working on separate papers that will specifically highlight the insights and perspectives of older adult participants and researchers involved in this study. These forthcoming publications will provide a comprehensive understanding of the research from multiple viewpoints. Thank you for raising this point, and we look forward to sharing the additional findings in the near future.
Reviewer 3 Report
Overall, I found the literature review to be limited. The scale was adequately explained . However, the experimental group's or intervention group's interaction with the population was not clearly stated. I was attempting to explain the article to a peer and could not clearly explain the intervention. Expand this section: "In the intervention group, students made weekly phone calls to older 65 adult participants based on reminiscence. In the control group, the weekly calls focused 66 on general topics including, diet and health, social activities, etc."
Overall weak explanation of the research, and limited literature review.
The quality of English writing was acceptable.
Author Response
- Overall, I found the literature review to be limited. The scale was adequately explained. However, the experimental group's or intervention group's interaction with the population was not clearly stated. I was attempting to explain the article to a peer and could not clearly explain the intervention. Expand this section: "In the intervention group, students made weekly phone calls to older 65 adult participants based on reminiscence. In the control group, the weekly calls focused 66 on general topics including, diet and health, social activities, etc."
Response: Thank you for this comment. We added details about the weekly calls for both intervention group and control groups on page 5.
- Overall weak explanation of the research, and limited literature review.
Response: Thank you for this comment. We misunderstood the length limit of the manuscript, which resulted in the literature review being shorter than anticipated. We have revised the manuscript to provide a more detailed literature review and ensured that the research and its related concepts are thoroughly explained.
Round 2
Reviewer 1 Report
After carefully reading the changes made by the authors and the accompanying letter, I believe that the quality of the article has been enhanced and it can be published. I congratulate the authors on their work.
Reviewer 2 Report
You have done an excellent job of responding to reviewers' suggestions. Well done.
Reviewer 3 Report
Accept with the revisions.